# Causal Inference and Shared Molecular Pathways in Crohn’s Disease, Celiac Disease, and Ankylosing Spondylitis: Integrative Mendelian Randomization and Transcriptomic Analysis

**DOI:** 10.3390/ijms26136451

**Published:** 2025-07-04

**Authors:** Ya Li, Shihao Xu, Mingzhu Zhang, Xin Yang, Zhengqiang Wei

**Affiliations:** First Affiliated Hospital of Chongqing Medical University, Chongqing 400042, China; 2019110189@stu.cqmu.edu.cn (Y.L.); 2021110226@stu.cqmu.edu.cn (S.X.); cqmuzhangmingzhu@163.com (M.Z.); yx774233326@163.com (X.Y.)

**Keywords:** Crohn’s disease, celiac disease, ankylosing spondylitis, Mendelian randomization, transcriptomic analysis, immune-mediated diseases

## Abstract

This study explored the causal and molecular overlap among Crohn’s disease (CD), celiac disease (CeD), and ankylosing spondylitis (AS). Bidirectional Mendelian randomization revealed significant causal associations between each disease pair. Transcriptomic analyses identified three consistently upregulated hub genes—P2RY8, ITGAL, and GPR65—across all conditions, which were validated in independent datasets and inflammatory cell models. Functional enrichment suggested these genes are involved in immune signaling and mucosal inflammation. Regulatory network and molecular docking analyses further highlighted Trichostatin A as a potential therapeutic agent. These findings reveal shared genetic and immune-related mechanisms, offering novel targets for cross-disease treatment strategies.

## 1. Introduction

Immune-mediated disorders, including Crohn’s disease (CD), ankylosing spondylitis (AS), and celiac disease (CeD), represent a significant global health burden due to their increasing prevalence, chronic nature, and impact on quality of life. These conditions, characterized by persistent inflammation, have complex pathogenesis and limited treatment options, underscoring the need for deeper mechanistic understanding and novel therapeutic strategies.

CD is a major form of inflammatory bowel disease characterized by transmural inflammation predominantly affecting the distal ileum and colon [1]. Its incidence has risen globally, with substantial morbidity resulting from complications such as strictures [2], fistulas [3], and abscesses [4]. AS is a chronic inflammatory disease of the axial skeleton [5], closely associated with HLA-B27 positivity [6], and manifests with back pain [5], stiffness, and progressive spinal fusion [7], often leading to disability [8]. CeD is an immune-mediated enteropathy triggered by gluten exposure [9], leading to villous atrophy [10] and impaired nutrient absorption [11], affecting nearly 1% of the global population with a rising incidence in recent decades [9]. While biologic therapies targeting TNF-α [12] and IL-17 [13] have improved symptom management in CD and AS [14], and a gluten-free diet remains the cornerstone of CeD management [15], none of these approaches fully prevent disease progression, and treatment responses vary considerably, highlighting the need for novel therapeutic strategies.

Emerging evidence suggests that CD, AS, and CeD share overlapping immunopathogenic mechanisms [16]. The concept of the gut–joint axis [17] highlights how dysregulated intestinal immunity and microbial dysbiosis can drive systemic inflammation and joint pathology. Impaired intestinal barrier integrity [18], microbial translocation [19], and dysregulated T cell responses (e.g., IL-23/Th17 axis) [20], as well as genetic variants such as IL23R [21] and ERAP1 [22], have been implicated in both gut and joint inflammation. In addition to genetic susceptibility, prior studies have also highlighted the potential roles of environmental triggers [23]. Klebsiella pneumoniae has been associated with the pathogenesis of AS and CD [24,25], and gluten exposure is the primary dietary driver in CeD [9]. These microbial and dietary antigens may act as upstream immune activators [26], possibly inducing autoimmunity through mechanisms such as molecular mimicry [27] or antigen presentation [28] in genetically predisposed individuals. This microbial connection is supported by several reviews that summarize how Klebsiella antigens may trigger inflammation through molecular mimicry [29], especially in HLA-B27-positive hosts [30]. Similarly, the gluten–CeD link has been extensively reviewed, demonstrating how gliadin peptides drive disease via HLA-DQ2/DQ8-restricted immune activation [31].

While these initiating factors differ among CD, AS, and CeD, their downstream immunological consequences may converge on common inflammatory circuits [32]. This convergence provides a compelling rationale for investigating shared molecular pathways and genetic drivers. Nevertheless, despite these shared immunological features and environmental links, most prior studies have examined these conditions in isolation or in pairwise comparisons—primarily between CD and AS, or CD and CeD [33]—while the possible connection between CeD and AS remains largely unexplored. Integrated analyses that combine genetic causal inference with transcriptomic data across all three diseases are even more limited. This lack of systematic investigation constrains our understanding of their converging mechanisms and hampers the development of targeted interventions for overlapping immune-mediated disorders.

To address these gaps, we conducted a systematic, integrative analysis combining bidirectional Mendelian randomization (MR) with transcriptomic profiling to investigate the causal relationships and shared molecular features among CD, AS, and CeD. By identifying key hub genes and pathways through this multi-layered approach, our study aims to provide novel insights into the common mechanisms underlying these immune-mediated diseases and highlight potential targets for therapeutic intervention.

## 2. Results

### 2.1. Causal Relationships Among CD, AS, and CeD Based on MR

The study workflow is outlined in Figure 1. To investigate the causal relationships among CD, AS, and CeD, we conducted MR analyses using instrumental variables that met genome-wide significance (*p* < 5 × 10^−8^). For each exposure–outcome pair, MR was performed using five complementary methods (IVW, WM, MR-Egger, simple mode, and weighted mode), and robustness was assessed through heterogeneity tests, MR-Egger regression for pleiotropy, and leave-one-out analysis.

After applying false discovery rate (FDR) correction, CD was found to have a significant causal effect on AS (IVW OR = 1.121, 95% CI = 1.066–1.178, FDR = 4.68 × 10^−5^), supported by WM analysis (FDR = 7.38 × 10^−6^). AS, in turn, showed a causal effect on CD (IVW OR = 2.189, 95% CI = 1.259–3.804, FDR = 0.0163). Bidirectional MR analysis indicated a causal relationship between CD and CeD (CD → CeD: IVW OR = 1.214, 95% CI = 1.037–1.422, FDR = 0.0314; CeD → CD: IVW OR = 1.112, 95% CI = 1.009–1.226, FDR = 0.0385). Lastly, AS was found to increase the risk of CeD (IVW OR = 2.241, 95% CI = 1.126–4.458, FDR = 0.0322), while CeD also exhibited a causal effect on AS (IVW OR = 1.033, 95% CI = 1.001–1.066, FDR = 0.0391).

For all tested causal relationships, no significant heterogeneity was detected in IVW (*p* > 0.05) or MR-Egger (*p* > 0.05) analyses, and MR-PRESSO showed no evidence of horizontal pleiotropy (*p* > 0.05). Leave-one-out analyses confirmed that no single SNP disproportionately influenced the overall estimates. These results, summarized in Figure 2, Figure 3 and Figure 4, and detailed in Appendix A, support robust causal relationships among CD, AS, and CeD, highlighting potential shared genetic mechanisms among these immune-mediated diseases.

### 2.2. Identification of DEGs

Building on the MR findings, we next explored shared molecular signatures across the three diseases by performing differential expression analysis based on transcriptomic datasets. In the CD dataset GSE207022, of the 1616 differentially expressed genes (DEGs), 1055 were upregulated and 561 were downregulated. In the CeD dataset GSE112102, there were 1299 DEGs, with 914 of them upregulated and 385 of them downregulated. In the AS dataset GSE25101, there were 914 DEGs, of which 503 were upregulated and 411 were downregulated. The expression patterns of DEGs are visualized using volcano plots in Figure 5A–C, and the differential expression for all DEGs through heatmaps is shown in Figure 5D–F.

### 2.3. WGCNA

We then applied weighted gene co-expression network analysis (WGCNA) to each dataset to identify co-expression modules and key regulatory genes. We conducted WGCNA on the CD dataset (GSE207022), the CeD dataset (GSE112102), and the AS dataset (GSE25101) to identify common genes influencing these diseases. The sample clustering dendrogram and scale-free topology model fitting and mean connectivity plots for the three diseases are shown in Appendix A. Using this method, four, four, and five co-expression modules were identified for each disease (Figure 6A–C), with the blue module in GSE207022, the turquoise module in GSE112102, and the blue module in GSE25101 demonstrating the strongest correlation with disease onset. Therefore, these three modules were chosen for additional analysis.

### 2.4. Selection and Validation of Hub Genes

We identified six candidate hub genes (P2RY8, RUNX3, S100A8, ITK, ITGAL, and GPR65) by intersecting differentially expressed genes (DEGs) with key WGCNA modules across the three diseases (Figure 7A). The expression levels of these genes were validated using independent GEO datasets: GSE95095 (CD), GSE164883 (CeD), and GSE73754 (AS). Among them, P2RY8, ITGAL, and GPR65 were consistently upregulated in CD, CeD, and AS (Figure 7B). Further validation was performed in inflammatory cell models, including an intestinal model (HT29 cells co-cultured with LPS) and a spinal model (MSCs co-cultured with TNF-α). The protein expression levels of the three genes in these models were consistent with the GEO validation results (Figure 7C). These three genes were thus confirmed as hub genes for subsequent analysis (Appendix A).

### 2.5. Diagnostic Performance of Key Genes in CD, CeD, and AS

The detected hub genes showed increased expression in samples from patients with CD, CeD, and AS compared to healthy individuals, suggesting their potential contribution to disease progression. In a separate validation cohort, the diagnostic performance of each gene was evaluated individually for CD, CeD, and AS. In addition, we constructed a combined predictive model by integrating the expression levels of P2RY8, ITGAL, and GPR65 in a multivariate logistic regression framework to assess their joint diagnostic power. Figure 8 displays the area under the receiver operating characteristic (ROC) curve (AUC) for each individual gene and the combined model, demonstrating that the combined model achieved higher AUC values across all three diseases.

### 2.6. Analysis of Hub Genes

We utilized the GeneMANIA database to analyze the co-expression network of these three genes, uncovering a complex protein-protein interaction (PPI) network (Figure 9A–C). GO analysis indicated that these genes are primarily associated with immune response, immune system regulation, and immune-related processes within the biological process category (Figure 9D). Regarding cellular components, the genes demonstrated enrichment in parts of the plasma membrane, the cell surface, and intrinsic membrane components (Figure 9E). Lastly, for molecular function, the genes were predominantly enriched in receptor signaling activity, molecular transduction activities, and transmembrane receptor signaling functions (Figure 9F). AlphaFold software (2024.10.30) was subsequently employed to model the structures of the proteins encoded by these genes and predict their interactions (Figure 9G). The predicted structures of the three proteins are presented in Appendix A.

### 2.7. GSEA Results of Hub Genes

This study delves into the roles of key hub genes in CD, CeD, and AS through GSEA analysis, identifying critical pathways linked to their high expression levels. High P2RY8 expression showed strong associations with key adaptive and innate immune pathways, particularly those governing IgA synthesis and NK cell cytotoxicity (Figure 10A–C). This suggests a central role for P2RY8 in coordinating mucosal immune defenses and maintaining intestinal homeostasis, underscoring its importance in shaping the immune microenvironment and regulating inflammation. Similarly, elevated ITGAL expression was tied to pathways essential for innate immune recognition and phagocytosis, highlighting its involvement in pathogen clearance and the activation of early immune responses (Figure 10D–F). Meanwhile, GPR65 overexpression corresponded to pathways influencing cell adhesion and antigen presentation, both crucial for immune surveillance and facilitating effective cellular interactions (Figure 10G–I). The nominal *p* values, NES, and FDR q values for GSE207022, GSE25101, and GSE112102 are presented in Appendix A. Collectively, these findings emphasize P2RY8, ITGAL, and GPR65 as pivotal regulators orchestrating immune responses in CD, CeD, and AS.

### 2.8. Drug Prediction and Molecular Docking

Finally, to explore potential therapeutic candidates targeting the shared hub genes, we performed in silico drug prediction and molecular docking analyses. Protein–drug interactions hold significant research value in uncovering the shared biological mechanisms of the three diseases. In this study, drugs targeting the ITGAL, GPR65, and P2RY8 genes were identified using the DSigDB database (Appendix A), with their intersections visualized via a Venn diagram (Figure 11A). A total of 49 drugs targeting ITGAL, 22 drugs targeting GPR65, and 10 drugs targeting P2RY8 were identified, among which Trichostatin A was found at the intersection, suggesting a potential link to molecular pathways associated with key genes. Subsequently, molecular docking analysis was performed to evaluate the binding affinity of Trichostatin A with key genes associated with the three diseases (Appendix A). A binding energy below −4.25 kcal/mol indicates a certain binding activity, below −5.0 kcal/mol represents moderate activity, and below −7.0 kcal/mol suggests strong binding affinity. The docking results showed that Trichostatin A binds to ITGAL (binding energy: −6.7 kcal/mol) via hydrogen bonds (VAL408, HIS586, and ALA356), hydrophobic interactions (VAL49 and ILE585), π-cation, π–π T-shaped interactions (HIS586), and alkyl interactions (PRO103 and CYS465) (Figure 11B). With GPR65 (binding energy: −8.1 kcal/mol), it forms hydrogen bonds (ASP172 and TYR269), hydrophobic interactions (HIS10), and π–alkyl interactions (TYR174) (Figure 11C). With P2RY8 (binding energy: −6.0 kcal/mol), it interacts via hydrogen bonds (ALA140), hydrophobic interactions (ARG135), π–σ interactions (VAL119 and TRP134), and π–alkyl interactions (TRP134) (Figure 11D). These findings suggest that Trichostatin A effectively binds to proteins encoded by the three key genes and may serve as a potential therapeutic candidate targeting these genes.

## 3. Discussion

In recent years, CD, CeD, and AS have been increasingly recognized for their overlapping immunopathogenic features, including abnormal immune cell activation [34], excessive pro-inflammatory cytokine release [35], and the breakdown of immune tolerance [10]. While previous studies have primarily focused on pairwise associations, a comprehensive understanding of their shared pathogenic mechanisms remains limited. Our study addresses this gap by integrating bidirectional MR and transcriptomic analyses, revealing robust causal relationships among CD, CeD, and AS, and identifying P2RY8, ITGAL, and GPR65 as key shared genes involved in immune regulation.

P2RY8 regulates B and T cell migration and function [36,37,38]; ITGAL (encoding CD11a) is critical for leukocyte adhesion [39] and T cell activation [40,41]; and GPR65 influences macrophage polarization [42] and T cell responses in acidic environments [43,44]. The aberrant expression or dysfunction of these genes may promote inflammation, suggesting a shared pathogenic mechanism across CD, CeD, and AS. The identification of these hub genes offers promising biomarkers for early diagnosis and potential therapeutic targets in immune-mediated diseases.

Given the pivotal roles of these genes in immune dysregulation, we further explored potential therapeutic strategies by focusing on Trichostatin A (TSA) [45], a well-characterized histone deacetylase inhibitor (HDACi) [46] known for its immunomodulatory properties. Molecular docking analyses demonstrated that TSA binds to P2RY8, ITGAL, and GPR65, revealing potential mechanisms linking epigenetic regulation to immune dysregulation. Specifically, TSA may modulate P2RY8 expression via the PI3K–AKT–mTOR pathway [47], influencing B cell activation [48] and antibody production [49]; downregulate ITGAL by inhibiting NF-κB [50] and JAK-STAT signaling [51], thereby reducing T cell adhesion and inflammatory responses [52], and enhance GPR65 expression by promoting histone acetylation [53], affecting macrophage polarization and fostering an anti-inflammatory environment. These findings highlight TSA as a promising candidate for drug repurposing in immune-mediated diseases, providing a mechanistic foundation for future studies on epigenetic therapies aimed at restoring immune homeostasis in complex inflammatory disorders such as CD, CeD, and AS.

To further explore the upstream regulatory landscape of the identified hub genes, we performed TF–gene and gene–miRNA interaction analyses. These supplementary analyses provide additional insights into the potential regulatory networks influencing P2RY8, ITGAL, and GPR65 in the context of immune dysregulation. Detailed results are presented in the Appendix A.

However, this study has several limitations. First, the GWAS datasets used in the MR analyses were predominantly derived from European populations, which, while ensuring homogeneity, may limit the generalizability of the findings to other ethnic groups due to differences in genetic architectures, linkage disequilibrium patterns, and environmental exposures. In addition, since the instrumental variables used in MR analyses were exclusively genetic, our causal inferences may not fully capture the influence of non-genetic factors such as microbial infections and dietary triggers, which are also known to contribute to disease development. Future studies should incorporate multi-ethnic GWAS datasets and explore integrative models that account for both genetic and environmental exposures. Furthermore, applying linkage disequilibrium score regression (LDSC) to estimate genome-wide genetic correlations among CD, CeD, and AS could provide valuable complementary insights into their shared genetic architecture. Second, the small sample size of the GSE112102 transcriptomic dataset may reduce statistical power and introduce potential biases, potentially affecting the reliability of our transcriptomic findings. Nevertheless, this dataset remains a valuable resource as one of the few publicly available datasets specifically focused on CeD. Future research should prioritize the inclusion of larger, more diverse transcriptomic datasets to validate and extend our results.

Furthermore, although we performed Western blot validation confirming the differential expression of P2RY8, ITGAL, and GPR65 in inflammatory conditions, additional functional studies are warranted to elucidate the precise roles of these genes in immune dysregulation. Future studies should also consider performing Fisher’s exact test to evaluate the statistical significance of shared genes across the diseases, which would provide complementary evidence for the robustness of our integrative findings. For CeD, the lack of appropriate cellular models remains a limitation, highlighting the need for patient-derived organoid systems in future studies. Finally, the rarity of patients diagnosed concurrently with CD, CeD, and AS poses challenges for direct clinical validation. Expanding large-scale biobanks and electronic health records (EHRs) could facilitate patient identification and provide a foundation for improving the diagnosis and management of these diseases.

Despite these limitations, our study provides important insights into the shared genetic mechanisms underlying CD, CeD, and AS and highlights key hub genes and potential therapeutic strategies that may inform the development of novel treatments for immune-mediated diseases. Notably, by incorporating CeD–AS associations, which are rarely explored in previous literature, our analysis fills a critical gap in the field. While prior research has predominantly focused on environmental triggers such as microbial infections [24] and dietary antigens, our study emphasizes genetic causality and transcriptomic dysregulation. By integrating Mendelian randomization, transcriptomic profiling, and molecular docking, we offer a complementary molecular perspective on disease pathogenesis. These approaches are not mutually exclusive but synergistic. Future research should incorporate multi-dimensional data—including environmental exposures, gut microbiota, and immunophenotyping—to further elucidate the interplay between genetic and non-genetic drivers. These efforts may ultimately lead to earlier diagnosis and more precise therapeutic interventions for complex immune-mediated disorders.

## 4. Materials and Methods

### 4.1. Data Source

The GWAS summary statistics for CD (2056 cases, 210,300 controls) were provided by FinnGen, the IGAS provided the data for AS (9069 cases, 1550 controls), and the European Bioinformatics Institute provided the data for CeD (4533 cases, 10,750 controls) (Appendix A). All of these studies involved European ancestry. The data were obtained from the MRC IEU Open GWAS, which encompasses 346.7 billion associations derived from 50,044 GWAS datasets. To investigate shared disease mechanisms, independent datasets from GEO were analyzed. These comprised GSE207022 for CD (125 patients, 23 controls), GSE112102 for celiac disease (12 patients, 12 controls), and GSE25101 for AS (16 patients, 16 controls) (Appendix A). The results of this analysis offer insights into common molecular patterns.

### 4.2. Selection of IVs

In this research, instrumental variables were extracted from the MR-Base database using the extract_instruments function, selecting SNPs with *p* < 5 × 10^−8^ for strong associations. Linkage disequilibrium (LD) clumping was applied to retain only independent SNPs [54]. The extract_outcome_data function ensured corresponding SNPs were present in outcome data, enabling causal inference. The harmonise_data function aligned exposure and outcome data, ensuring consistency in allele direction and minimizing bias. Additionally, the MR-PRESSO method was employed to detect and correct for potential horizontal pleiotropy, ensuring the validity of instrumental variables.

### 4.3. MR Analyses

We performed standard MR analysis using the MR function from the TwoSampleMR package in R 4.4.1 [55], based on exposure and outcome data. Causal effects were estimated through multiple methods, including inverse variance weighting (IVW), Egger regression, and the weighted median method. To account for multiple testing across bidirectional MR analyses, we applied false discovery rate (FDR) correction using the Benjamini–Hochberg method. Results were interpreted based on FDR-adjusted *p*-values. To enhance interpretation, the results were converted to odds ratios (OR).

### 4.4. Sensitivity Analyses

The assessment of heterogeneity was conducted using the mr_heterogeneity function, which employs Cochran’s Q statistic and *p*-value to evaluate the impact of single-nucleotide polymorphisms (SNPs) on the outcome, thereby identifying variability in instrumental variable effects. The mr_pleiotropy_test function, through the application of Egger regression, estimated the intercept with the objective of assessing horizontal pleiotropy. A non-zero intercept indicates that the SNPs may exert an influence on the outcomes via exposure-independent pathways, thereby suggesting the potential for pleiotropy. A sensitivity analysis was conducted using the mr_leaveoneout function, whereby each SNP was removed in turn to allow for reassessment of the causal effect. The consistency of results after the removal of any single nucleotide polymorphism (SNP) indicates the robustness of the findings, confirming that no single SNP unduly influences the overall effect estimate.

### 4.5. Differentially Expressed Genes

We downloaded the raw count data and normalized it using the TPM (Transcripts Per Million) method to adjust for variations in sequencing depth and gene length. Differential expression analysis of the CeD (GSE112102), AS (GSE25101), and CD (GSE207022) datasets was then performed using the limma package in R 4.4.1. During this analysis, probe sets without gene symbols were excluded, and for genes with multiple probe sets, the median expression value was used for further analysis. DEGs were defined by an adjusted *p*-value below 0.05 and a fold change exceeding 1.5. To visualize the DEGs in each dataset, volcano plots were generated. Overlapping DEGs across all datasets were identified as core shared genes and extracted for downstream functional enrichment analysis.

### 4.6. Construction of Co-Expression Networks Using WGCNA

The Weighted Gene Co-expression Network Analysis (WGCNA) method was utilized to construct co-expression networks for the datasets GSE112102, GSE207022, and GSE25101. Prior to network construction, a data pre-processing procedure was applied to ensure data quality and reliability. Specifically, genes with low expression were filtered out by retaining only those with a standard deviation greater than 0.5 across samples, ensuring sufficient variability for meaningful analysis. Quantile normalization was then performed across samples to adjust for systematic technical differences and improve data comparability. After preprocessing, a soft-thresholding power was selected to establish scale-free networks. The optimal soft-thresholding powers were β = 18 for CD, β = 8 for AS, and β = 15 for CeD, based on the criterion that the scale-free topology fit index exceeded 0.85 while maintaining reasonable mean connectivity. The interconnectivity between genes was quantified using an adjacency matrix, which was subsequently transformed into a topological overlap matrix (TOM) to strengthen connectivity. Hierarchical clustering of the TOM identified distinct gene modules, and module eigengenes were calculated to assess their correlation with disease occurrence. Significant modules were selected, and hub genes consistently identified across multiple datasets and highly relevant to disease processes were retained for further analysis.

### 4.7. Confirmation of Hub Gene Expression

Hub genes were identified by integrating WGCNA module analysis and differential expression analysis (DEGs). Specifically, for each disease, co-expression modules were identified using WGCNA, and DEGs were identified separately. The intersection of module genes and DEGs was defined as hub genes, representing genes that are both co-expressed and differentially expressed across immune-mediated diseases. The expression levels of the hub genes were validated using three independent datasets: GSE95095 for CD (12 normal and 24 CD intestinal samples), GSE73754 for AS (21 normal and 51 AS peripheral blood samples), and GSE164883 for CeD (21 normal and 25 CeD intestinal samples). Independent *t*-tests were used for comparisons, with statistical significance set at *p* < 0.05.

### 4.8. Cell Culture

HT29 cells, used for constructing the CD cell model, were cultured in RPMI 1640 medium supplemented with 10% fetal bovine serum (FBS), 100 U/mL penicillin, and 100 U/mL streptomycin. MSCs (Procell, CP-H166, Wuhan, China), used for constructing the AS cell model, were cultured in complete human bone marrow mesenchymal stem cell medium (Procell, CM-H166). All cells were maintained in a humidified incubator at 37 °C with 5% CO_2_ to ensure optimal growth conditions.

### 4.9. Western Blot

For protein extraction, cells were initially rinsed with PBS. Next, 100 μL of lysis buffer was applied, and gentle pipetting facilitated cell disruption. The lysate was then centrifuged at 14,000× *g* for 5 min at 4 °C, and the supernatant was carefully collected for downstream analysis. Protein concentration was assessed via the BCA assay, adjusted to 40 μg per 20 μL, and subsequently denatured in a metal bath at 100 °C for 5 min. Proteins were resolved using SDS-PAGE with electrophoresis lasting 70 min, followed by a 90-min membrane transfer step. Post-transfer, the membrane underwent a 15-min blocking step before being rinsed three times with TBST. The primary antibody (1:1000 dilution) was incubated at 4 °C overnight on a shaker. After removal of the primary antibody, the membrane was washed thrice with TBST, then incubated with the secondary antibody at room temperature for 2 h. Lastly, protein bands were detected using BeyoECL Star reagent(Cat. No. P0018AS, Beyotime Biotechnology, Shanghai, China) and visualized via a chemiluminescence imaging system.

### 4.10. GeneMANIA Analysis

We used GeneMANIA “http://genemania.org (accessed on 3 November 2024)” to construct an interaction network for the target genes, which was automatically extended with functionally similar or interacting genes. Key nodes and clusters were identified and cross-referenced with the literature, with further analysis using other bioinformatics tools to ensure reliability.

### 4.11. Protein Structure Prediction Using AlphaFold

Wild-type protein models were obtained from the AlphaFold database using UniProt “https://www.uniprot.org/ (accessed on 15 March 2025)” access codes [56], and interaction schemes were predicted using the AlphaFold Server. Using AlphaFold 3 models, the results were visualized using 3D Viewer and Pymol software for detailed structural analysis.

### 4.12. Receiver Operating Characteristic Curves

To assess the predictive performance of the hub genes, we used the pROC package in R to generate ROC curves and calculate the area under the curve (AUC). AUC values closer to 1 indicate higher diagnostic accuracy, while values near 0.5 suggest no predictive value. We used a 95% confidence interval based on 2000 bootstrap replicates. Additionally, a combined predictive model was built by including the expression levels of P2RY8, ITGAL, and GPR65 as predictors in a multivariate logistic regression, with disease status as the outcome. The combined model’s predicted probabilities were used to generate ROC curves and AUC values, allowing comparison with individual gene performance.

### 4.13. Gene Set Enrichment Analysis (GSEA)

The GSEA software (version 3.0) was employed to categorize the samples into high- and low-expression groups. The c2.cp.kegg.v7.4 gene set [57] was acquired to evaluate the pathways. The gene sets were defined with a range of sizes between 5 and 5000, with 1000 permutations, with a threshold for statistical significance set at *p* < 0.05.

### 4.14. Transcription Factor–Gene Interactions

NetworkAnalyst 3.0 “https://www.networkanalyst.ca/ (accessed on 7 March 2025)” was employed for the analysis of interactions between common genes and transcription factors (TFs) utilizing ENCODE ChIP-seq data. Interactions with peak intensity signals lower than 500 and regulatory potential scores less than 1 were considered in the analysis. The resulting TF–gene regulatory network was visualized using the Cytoscape (version 3.9.1) software.

### 4.15. Gene–miRNA Interactions

The data on miRNA–gene interactions were obtained from TarBase v9.0, which provides experimentally validated interactions. The miRNA-gene regulatory network was constructed and visualized using Cytoscape, with the objective of identifying key miRNA regulators in order to improve the interpretability of gene regulation.

### 4.16. Molecular Docking

For the target protein, the AlphaFold-predicted protein structure was processed using Pymol-2.1.0, including hydrogen removal, amino acid modification, energy optimization, and force field parameter adjustment. The protein was then subjected to hydrogen addition and charge processing using AutoDock Tools-1.5.6 http://vina.scripps.edu/ (accessed on 14 March 2025) and saved in pdbqt format. The ligand structure was downloaded from the PubChem database “https://pubchem.ncbi.nlm.nih.gov/” (accessed on 14 March 2025). Finally, molecular docking was performed between the target structure and the active compound structure using Vina within the PyRx software https://pyrx.sourceforge.io/ (accessed on 14 March 2025). The Affinity (kcal/mol) value represents the binding capacity between the two molecules; a lower value indicates a more stable ligand–receptor interaction. The docking results were visualized using Discovery Studio 2019.

## 5. Conclusions

This study provides integrative genetic and transcriptomic evidence for shared immune mechanisms linking Crohn’s disease, celiac disease, and ankylosing spondylitis. The identified hub genes and potential therapeutic targets may inform future cross-disease diagnostic and treatment strategies.

## Figures and Tables

**Figure 1 ijms-26-06451-f001:**
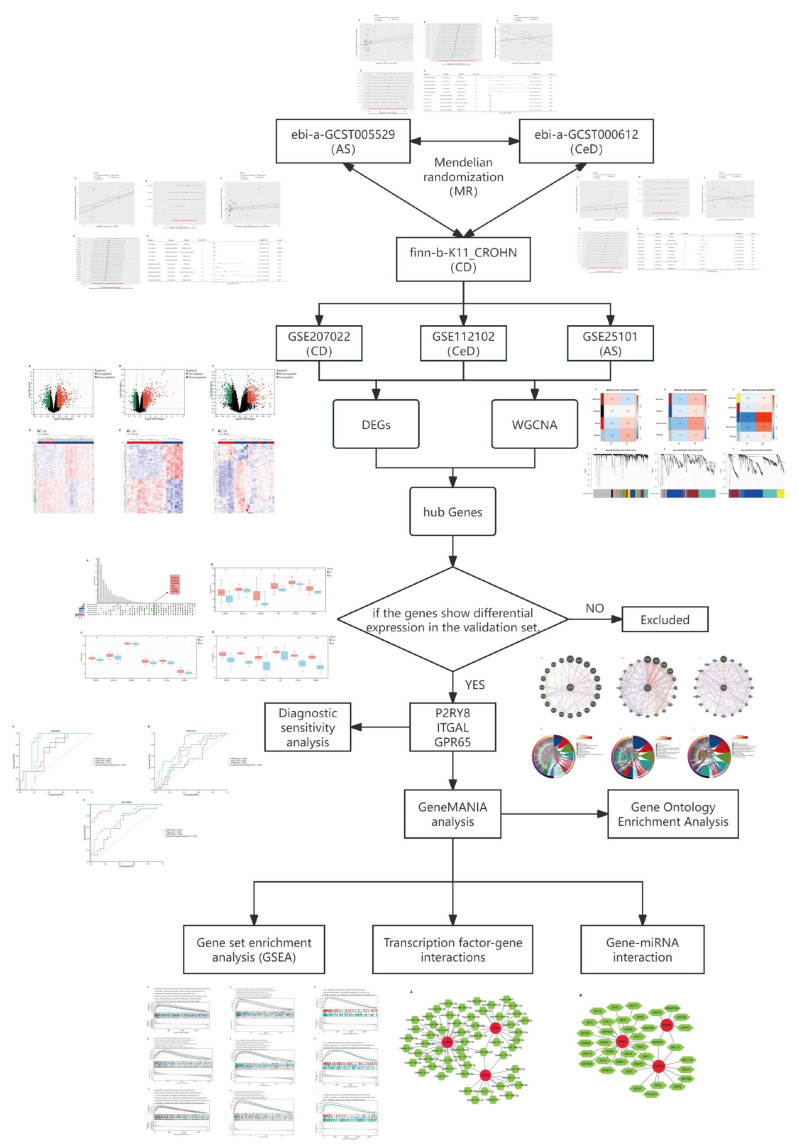
The flowchart illustrating the complete study design.

**Figure 2 ijms-26-06451-f002:**
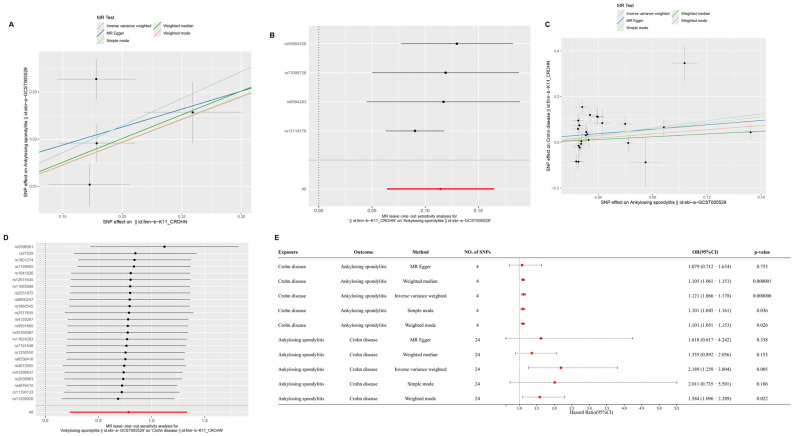
Bidirectional Mendelian randomization (MR) analysis between Crohn’s disease (CD) and ankylosing spondylitis (AS). (**A**) Scatterplot of SNP effect sizes for CD (exposure) and AS (outcome) using five MR methods: inverse variance weighted (IVW), MR-Egger, weighted median, simple mode, and weighted mode. (**B**) MR leave-one-out sensitivity test for CD on AS. (**C**) Scatterplot of SNP effect sizes for AS (exposure) and CD (outcome) using the same MR methods. (**D**) MR leave-one-out sensitivity test for AS on CD. (**E**) Forest plot of odds ratios (ORs) and 95% confidence intervals (CIs) for the causal relationships between CD and AS using different MR methods. Four SNPs were used for CD and 24 SNPs for AS.

**Figure 3 ijms-26-06451-f003:**
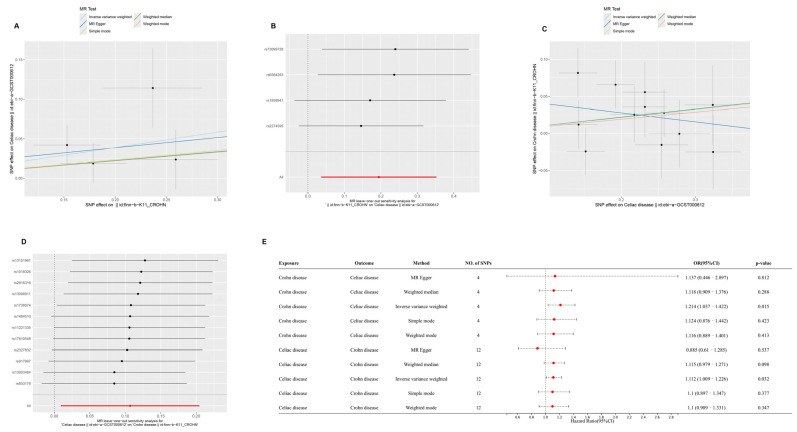
Bidirectional Mendelian randomization (MR) analysis between Crohn’s disease (CD) and celiac disease (CeD). (**A**) Scatterplot of SNP effect sizes for CD (exposure) and CeD (outcome) using five MR methods: inverse variance weighted (IVW), MR-Egger, weighted median, simple mode, and weighted mode. (**B**) MR leave-one-out sensitivity test for CD on CeD. (**C**) Scatterplot of SNP effect sizes for CeD (exposure) and CD (outcome) using the same MR methods. (**D**) MR leave-one-out sensitivity test for CeD on CD. (**E**) Forest plot of odds ratios (ORs) and 95% confidence intervals (CIs) for the causal relationships between CD and CeD using different MR methods. Four SNPs were used for CD and 12 SNPs for CeD.

**Figure 4 ijms-26-06451-f004:**
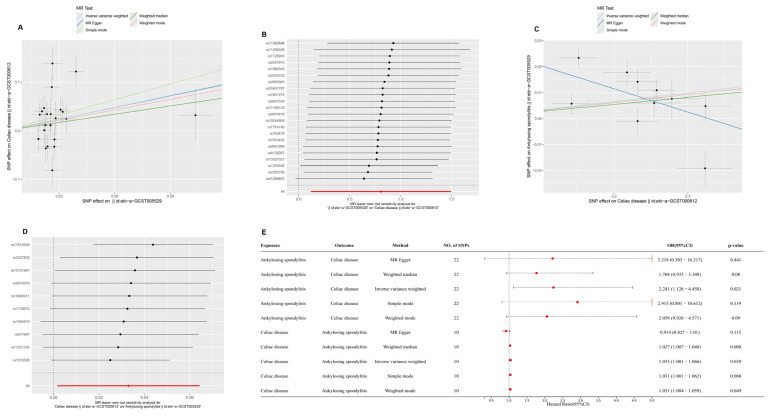
Bidirectional Mendelian randomization (MR) analysis between ankylosing spondylitis (AS) and celiac disease (CeD). (**A**) Scatterplot of SNP effect sizes for AS (exposure) and CeD (outcome) using five MR methods: inverse variance weighted (IVW), MR-Egger, weighted median, simple mode, and weighted mode. (**B**) MR leave-one-out sensitivity test for AS on CeD. (**C**) Scatterplot of SNP effect sizes for CeD (exposure) and AS (outcome) using the same MR methods. (**D**) MR leave-one-out sensitivity test for CeD on AS. (**E**) Forest plot of odds ratios (ORs) and 95% confidence intervals (CIs) for the causal relationships between AS and CeD using different MR methods. Twenty-two SNPs were used for AS and 10 SNPs for CeD.

**Figure 5 ijms-26-06451-f005:**
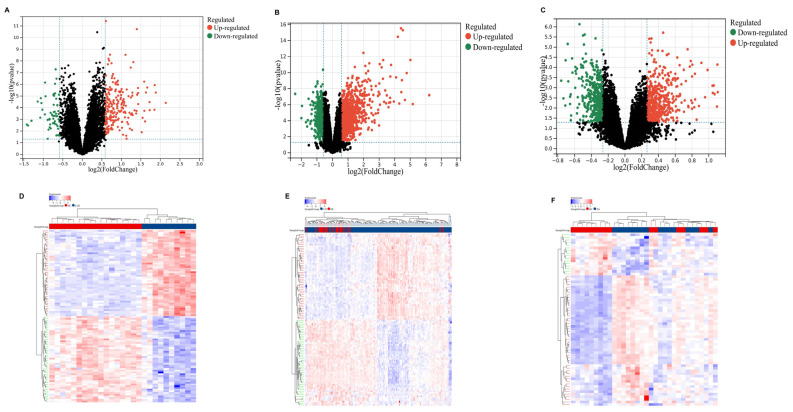
Differentially expressed genes (DEGs) in Crohn’s disease (CD), celiac disease (CeD), and ankylosing spondylitis (AS). (**A**–**C**) Volcano plots of DEGs identified in (**A**) CD, (**B**) CeD, and (**C**) AS. Red points represent upregulated genes, and green points represent downregulated genes (adjusted *p*-value < 0.05, fold change > 1.5). (**D**–**F**) Heatmaps showing the expression levels of DEGs in (**D**) CD, (**E**) CeD, and (**F**) AS. Each row represents a gene, and each column represents a sample. Red indicates higher expression, and blue indicates lower expression, with clustering of samples based on gene expression profiles.

**Figure 6 ijms-26-06451-f006:**
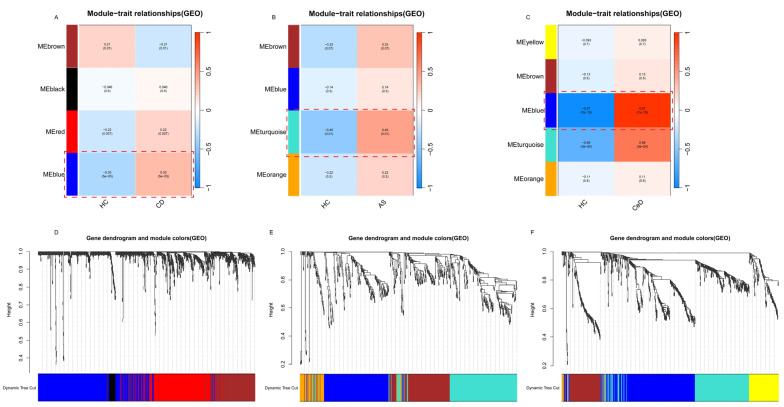
Weighted gene co-expression network analysis (WGCNA) modules correlated with disease traits in Crohn’s disease (CD), ankylosing spondylitis (AS), and celiac disease (CeD). (**A**–**C**) Module–trait heatmaps showing the correlations between gene co-expression modules and disease traits in (**A**) CD, (**B**) AS, and (**C**) CeD. Colors indicate the correlation strength (red: positive correlation; blue: negative correlation). (**D**–**F**) Dendrograms of genes clustered into modules based on co-expression patterns in (**D**) CD, (**E**) AS, and (**F**) CeD datasets. Each color below the dendrogram represents a module, and dynamic tree cut was used to identify the modules.

**Figure 7 ijms-26-06451-f007:**
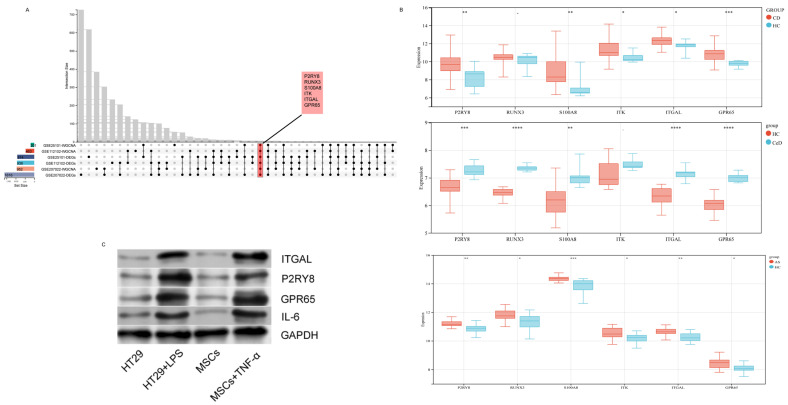
Identification and validation of overlapping hub genes in Crohn’s disease (CD), ankylosing spondylitis (AS), and celiac disease (CeD). (**A**) Upset plot showing the overlap of differentially expressed genes (DEGs) and key weighted gene co-expression network analysis (WGCNA) modules across CD, AS, and CeD datasets. Six hub genes (P2RY8, RUNX3, S100A8, ITK, ITGAL, and GPR65) were identified as shared among all three diseases. (**B**) Boxplots of the expression levels of the six hub genes in CD, AS, and CeD compared to healthy controls (HC). Red represents disease groups (CD, AS, or CeD), and blue represents healthy controls. *p* values indicate the statistical significance of differences in expression levels (* *p* < 0.05, ** *p* < 0.01, *** *p* < 0.001, **** *p* < 0.0001). (**C**) Western blot analysis of protein ITGAL, P2RY8, GPR65, and IL-6 of HT29 and MSCs in different conditions.

**Figure 8 ijms-26-06451-f008:**
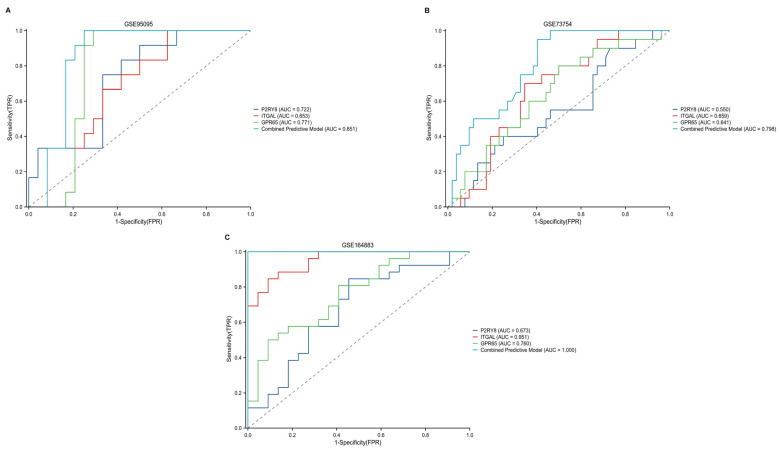
Receiver operating characteristic (ROC) curves for the diagnostic performance of hub genes in Crohn’s disease (CD), ankylosing spondylitis (AS), and celiac disease (CeD). (**A**) ROC curve for P2RY8, ITGAL, and GPR65 in the GSE95095 dataset (CD), showing the area under the curve (AUC) values for each gene and the combined predictive model. (**B**) ROC curve for the same hub genes in the GSE73754 dataset (AS), with AUC values for individual genes and the combined model. (**C**) ROC curve for hub gene performance in the GSE164883 dataset (CeD), with AUC values for each gene and the combined predictive model. The combined predictive model demonstrated higher diagnostic accuracy across all datasets.

**Figure 9 ijms-26-06451-f009:**
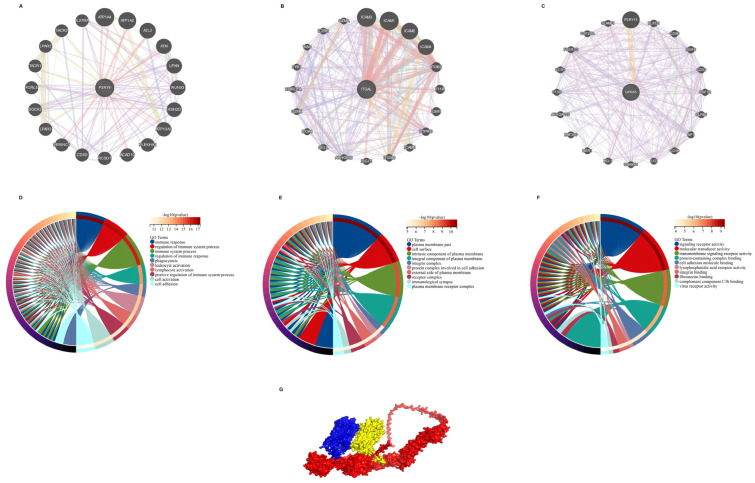
Protein–protein interaction (PPI) networks and gene ontology (GO) analysis of hub genes. (**A**–C) PPI networks for (**A**) P2RY8, (**B**) ITGAL, and (**C**) GPR65, showing interactions with other proteins based on GeneMANIA analysis. Each node represents a protein, and edges represent interactions, with colors indicating different types of functional relationships. (**D**–**F**) GO enrichment analysis of the hub genes’ biological processes, cellular components, and molecular functions. (**D**) Biological processes, highlighting immune response, regulation of immune system processes, and phagocytosis. (**E**) Cellular components, showing enrichment in plasma membrane components and the immunological synapse. (**F**) Molecular functions, focusing on receptor activity and signaling pathways. The length of each arc represents the significance level (*p*-value). (**G**) Schematic representation of the predicted interaction network among P2RY8, ITGAL, and GPR65, visualized using Pymol.

**Figure 10 ijms-26-06451-f010:**
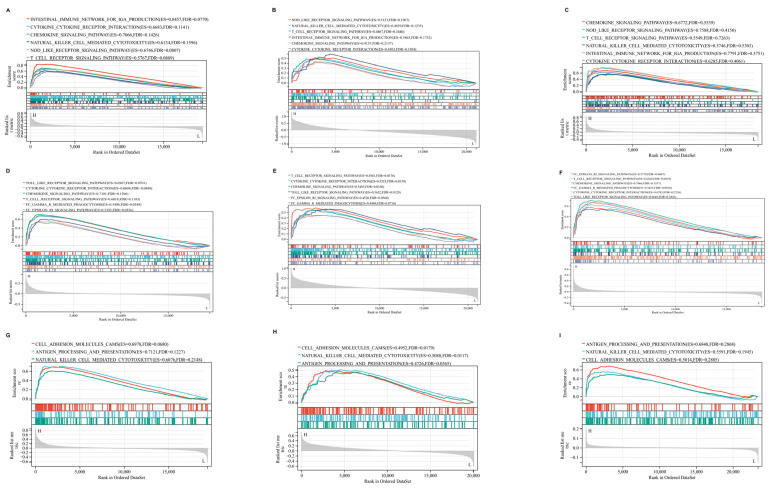
Gene set enrichment analysis (GSEA) of hub genes in Crohn’s disease (CD), ankylosing spondylitis (AS), and celiac disease (CeD). (**A**–**C**) Results of GSEA enrichment analyses of P2RY8 in CD, AS, and CeD datasets. (**D**–**F**) Results of GSEA enrichment analyses of ITGAL in CD, AS, and CeD datasets. (**G**–**I**) Results of GSEA enrichment analyses of GPR65 in CD, AS, and CeD datasets. The enrichment score and ranked gene list are shown, with higher scores indicating stronger pathway enrichment.

**Figure 11 ijms-26-06451-f011:**
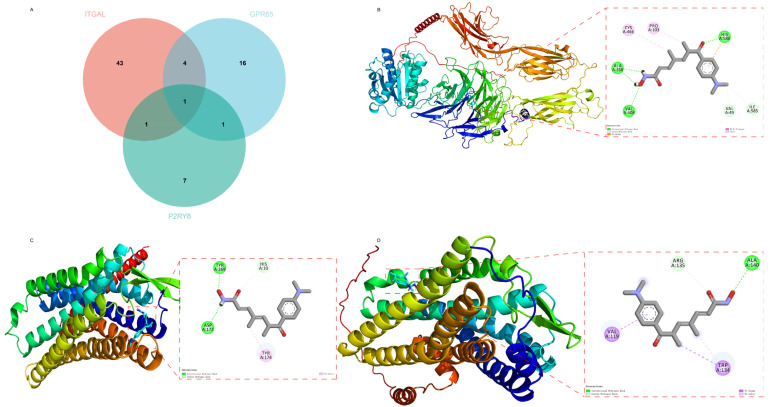
Drug prediction and molecular docking. (**A**) Venn diagram of drug predictions for the three key genes. (**B**) Structural diagram of molecular docking between the drug and ITGAL. (**C**) Structural diagram of molecular docking between the drug and GPR65. (**D**) Structural diagram of molecular docking between the drug and P2RY8.

## Data Availability

The original contributions presented in this study are included in the article/Appendix A. Further inquiries can be directed to the corresponding author.

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
