# Peer review of "Causal Inference and Shared Molecular Pathways in Crohn’s Disease, Celiac Disease, and Ankylosing Spondylitis: Integrative Mendelian Randomization and Transcriptomic Analysis"

_ijms, 2025, doi:10.3390/ijms26136451_

Round 1
Reviewer 1 Report
Comments and Suggestions for Authors
In this article by Li et al., the authors tried to evaluate the causal relationship and molecular overlap among three different diseases: Crohn’s disease, celiac disease, and ankylosing spondylitis. One of the major drawbacks of the manuscript is that the figures are of very poor quality and difficult to understand. The authors should pay more attention to this. Some figures are also getting cropped (Figure 2). Although there are limited abbreviations present, the authors should introduce any abbreviations when they are first mentioned. It will help the readers. The manuscript needs to be more organized to be able to proceed further.
Author Response
Comment 1:
One of the major drawbacks of the manuscript is that the figures are of very poor quality and difficult to understand. Some figures are also getting cropped (Figure 2).
Response:
Thank you for your comment regarding the figure quality. We sincerely apologize for any inconvenience caused by the previous low-resolution images. In this revised submission, we have carefully enhanced the resolution of all figures and ensured they meet the required publication standards. We appreciate your understanding and hope that the revised figures now provide clearer visual information for readers.
Comment 2:
Although there are limited abbreviations present, the authors should introduce any abbreviations when they are first mentioned.
Response:
We thank the reviewer for this helpful suggestion. We have revised the manuscript to ensure that all abbreviations are clearly introduced upon first mention. We appreciate the reviewer’s attention to detail.
Comment 3:
The manuscript needs to be more organized to be able to proceed further.
Response:
Thank you for this helpful comment. We have carefully revised the manuscript to improve the overall organization, logical flow, and clarity. Specifically, we have:
- Refined the Introduction by providing a clearer overview of previous studies on the shared mechanisms of CD, CeD, and AS, and by emphasizing the novelty and objectives of our study.
- Improved the Discussion section by systematically comparing our findings with existing literature, clarifying the biological significance of the identified hub genes (P2RY8, ITGAL, and GPR65), and discussing the potential therapeutic role of Trichostatin A in immune-mediated diseases.
- Added future directions in the Discussion, including plans to perform Fisher’s exact test to assess the statistical significance of shared genes and LDSC analysis to evaluate genome-wide genetic correlations across CD, CeD, and AS.
- Acknowledged key limitations, such as the reliance on European GWAS datasets and the limited sample size of the CeD transcriptomic dataset, and proposed strategies for addressing these issues in future research.
- Clarified methodological details in the revised Methods section, including WGCNA parameters, hub gene selection criteria, and the construction of the combined predictive model.
We hope these revisions have improved the clarity, structure, and readability of the manuscript, and we sincerely appreciate the reviewer’s suggestions.
Reviewer 2 Report
Comments and Suggestions for Authors
In this study, the authors explored molecular commonalities between three autoimmune diseases, namely CD, CeD and AS. They first conducted bidirectional MR analyses to assess causal relationships, and consequently identified shared genes through transcriptomic analysis of publicly available datasets. The authors’ attempt to validate their findings in an experimental setting is appreciated. Nevertheless, the authors have omitted an abundance of important details in both the methodology and results, undermining the significance of their findings. Statistics are not described in the core results section (2.3 - 2.5), while additional in silico investigations seem unnecessary (e.g., TF-gene interactions, gene-miRNA interactions).
General notes
- I thoroughly checked all provided files, however I was unable to retrieve all relevant supplementary files apart from the blots.
- The quality of the submitted figures must be enhanced prior to additional considerations.
Major comments
- In MR, the authors performed several different bidirectional MR analyses (e.g., CD was both set as exposure and as outcome), however no correction for multiple testing was performed. The authors must re-evaluate their statistical significance through p-value correction methods.
- It would be interesting for the authors to examine the statistical significance of shared genes through Fisher’s exact test, and a global genetic correlation analysis through LDSC. This would further provide a holistic image of possible commonalities between all 3 diseases.
- In WGCNA, I have several concerns regarding the validity of the applied methodology
a) How did the authors select the soft threshold? What were the criteria?
b) The absence of an analytical number of included transcripts is spurious. The authors do mention at lines 588-591 their pre-processing steps, however these are vaguely described. What kind of normalization and filtering of low-expression genes were performed?
c) In WGCNA, genes in the grey module are considered as unassigned, meaning they don’t really fit anywhere else and are not taken into further consideration for follow-up analyses.
d) How did the authors identify hub genes for each module? What were the parameters used? By default, WGCNA considers hub genes as those genes with the highest connectivity in each module (except the grey module). This does not seem to be the case here. If not, this should be clarified in the manuscript.
4. Fig. 8. How do the authors explain the “combined predictive model”? This is not even explained in the manuscript.
Author Response
Major Comment 1:
In MR, the authors performed several different bidirectional MR analyses (e.g., CD was both set as exposure and as outcome), however no correction for multiple testing was performed. The authors must re-evaluate their statistical significance through p-value correction methods.
Response:
We thank the reviewer for this insightful comment. Following the recommendation, we have applied False Discovery Rate (FDR) correction using the Benjamini-Hochberg method to account for multiple testing across all bidirectional Mendelian randomization analyses. The corrected p-values have been incorporated into the revised manuscript (Lines:70-78, and Supplementary file 3 ). We also clearly state in the Methods section (Lines:593-595) that FDR correction was used, and have revised the Results section to indicate which associations remain statistically significant after correction. We appreciate the reviewer’s valuable suggestion, which has strengthened the rigor and robustness of our study.
Major Comment 2:
It would be interesting for the authors to examine the statistical significance of shared genes through Fisher’s exact test, and a global genetic correlation analysis through LDSC. This would further provide a holistic image of possible commonalities between all 3 diseases.
Response:
Thank you for this valuable suggestion. We agree that examining the statistical significance of shared genes using Fisher’s exact test and exploring global genetic correlations through linkage disequilibrium score regression (LDSC) would provide a more comprehensive understanding of the commonalities among Crohn’s disease, celiac disease, and ankylosing spondylitis.
While these analyses were not included in the current study, as our focus was on integrating Mendelian randomization, transcriptomic analysis, and molecular docking to identify causal relationships and shared molecular mechanisms, we fully recognize the value of these additional approaches. We plan to incorporate Fisher’s exact test and LDSC in our future research to further validate and complement our findings. We have acknowledged this point and the need for these analyses in the revised Discussion section(lines:539-542;lines:551-553).
Major Comment 3:
In WGCNA, I have several concerns regarding the validity of the applied methodology
a) How did the authors select the soft threshold? What were the criteria?
Response:
We thank the reviewer for this important question. We selected the optimal soft-thresholding power by using the pickSoftThreshold() function from the WGCNA package, which identifies the power parameter that best satisfies the scale-free topology criterion. Specifically, we chose the lowest power value at which the scale-free topology fit index (R²) exceeded 0.85 while maintaining a reasonable mean connectivity. The results, including the scale-free topology model fit and mean connectivity plots, have been provided in the Supplementary Materials (Supplementary file 4).
b) The absence of an analytical number of included transcripts is spurious. The authors do mention at lines 588-591 their pre-processing steps, however these are vaguely described. What kind of normalization and filtering of low-expression genes were performed?
Response:
Thank you for the comment. Prior to network construction, we applied a data pre-processing procedure to ensure the quality and reliability of the gene expression matrix. Specifically, we filtered out low-expression genes by retaining only those with a standard deviation greater than 0.5 across samples, as these genes demonstrate sufficient variation for meaningful analysis. Additionally, we performed quantile normalization across samples to adjust for systematic technical differences and improve comparability between arrays. These steps were essential to obtain robust co-expression modules and to reduce noise in the subsequent analyses. We have clarified this data processing procedure in the revised Methods section (lines:623-628).
c) In WGCNA, genes in the grey module are considered as unassigned, meaning they don’t really fit anywhere else and are not taken into further consideration for follow-up analyses.
Response:
Thank you for your valuable comment. We sincerely apologize for the confusion. In the original figure, we unintentionally used grey as a color for one of the actual modules, which might have led to a misunderstanding that we were referring to the "grey module" (unassigned genes) in WGCNA. However, these genes were not the default unassigned genes, but were part of a valid co-expression module, and they were included in the downstream analyses accordingly. To avoid confusion, we have re-run the code and reassigned module colors, ensuring that grey is not used as a module color in the revised figures. We hope this clarification resolves the concern.
d) How did the authors identify hub genes for each module? What were the parameters used? By default, WGCNA considers hub genes as those genes with the highest connectivity in each module (except the grey module). This does not seem to be the case here. If not, this should be clarified in the manuscript.
Response:
We thank the reviewer for this important comment. In the revised manuscript, we have clarified the procedure for identifying hub genes. Specifically, we did not rely solely on intramodular connectivity (kWithin) as the basis for hub gene selection. For each of the three diseases (CD, CeD, and AS), we performed WGCNA analysis to identify co-expression modules and differential expression analysis (DEGs) to identify differentially expressed genes. This resulted in six gene sets (three modules and three DEGs). We then intersected these six gene sets, and the overlapping genes were defined as hub genes for further analysis. This integrative approach prioritizes genes that are both co-expressed within modules and consistently differentially expressed across multiple immune-mediated diseases, providing a complementary perspective beyond intramodular connectivity alone. The description of this procedure has been added to the revised Methods section (lines:110-113;639-643). We hope this clarification addresses the reviewer’s concern.
Major Comment 4:
Fig. 8. How do the authors explain the “combined predictive model”? This is not even explained in the manuscript.
Response:
We thank the reviewer for this important comment and for helping us identify this oversight. In the revised manuscript, we have clarified the construction of the "combined predictive model," which refers to a multivariate logistic regression model built to evaluate the joint diagnostic performance of the identified hub genes (P2RY8, ITGAL, and GPR65). The model uses the expression levels of these genes as independent variables and disease status (CD, CeD, or AS versus healthy controls) as the dependent variable. The predicted probabilities were used to generate ROC curves, and the area under the curve (AUC) was calculated to assess diagnostic accuracy. The combined model demonstrated higher AUC values across all datasets compared to predictions based on individual genes, indicating that integrating multiple gene features enhances diagnostic performance (see Figure 8A–C). We have added a detailed description of this model and its evaluation process in the revised Methods section (lines:685-689) and Results section (lines:123-128). We sincerely appreciate the reviewer’s helpful suggestion, which has allowed us to improve the clarity and completeness of the manuscript.
General notes 1:
I thoroughly checked all provided files, however I was unable to retrieve all relevant supplementary files apart from the blots.
Response:
Thank you for your careful review and for bringing this to our attention. We sincerely apologize for any inconvenience caused by the missing supplementary files. In this revised submission, all supplementary materials have been carefully compiled and included in a single ZIP archive for your convenience. We are not sure why the files were not accessible previously, but we have double-checked to ensure that all relevant data are now properly uploaded. Please let us know if there are any further issues or if additional clarifications are needed.
General notes 2:
The quality of the submitted figures must be enhanced prior to additional considerations.
Response:
Thank you for your comment regarding the figure quality. We sincerely apologize for any inconvenience caused by the previous low-resolution images. In this revised submission, we have carefully enhanced the resolution of all figures and ensured they meet the required publication standards. We appreciate your understanding and hope that the revised figures now provide clearer visual information for readers.
Suggestions 1:
while additional in silico investigations seem unnecessary (e.g., TF-gene interactions, gene-miRNA interactions).
Response:
We thank the reviewer for this valuable comment. We agree that a clear focus on core results is important and have revised the manuscript accordingly. However, we believe that the TF–gene and gene–miRNA interaction analyses provide meaningful insights by elucidating the upstream regulatory networks of the identified hub genes. Given the complex immune regulation underlying CD, AS, and CeD, integrating transcription factors and miRNAs helps to better understand how these hub genes may be controlled in the broader immune landscape, offering potential directions for future experimental validation and therapeutic target development.
To address the reviewer’s concern, we have moved the detailed results of these analyses to the Supplementary file 10, and only briefly mentioned them in the Discussion section(lines:529-533) to avoid distracting from the core findings. We have also added clarifications in the Methods section to ensure transparency. We hope this balanced approach clarifies the purpose of these analyses while maintaining a clear focus on the primary results of the study.
Reviewer 3 Report
Comments and Suggestions for Authors
This manuscript titled "Causal Inference and Shared Molecular Pathways in Crohn's Disease, Celiac Disease, and Ankylosing Spondylitis: Integrative Mendelian Randomization and Transcriptomic Analysis" by Ya Li et al. presents an integrative analysis exploring the causal and molecular overlap among Crohn's disease (CD), celiac disease (CeD), and ankylosing spondylitis (AS) using Mendelian randomization and transcriptomic approaches. While the study addresses an important area of research and provides novel insights into shared mechanisms among these immune-mediated diseases, several major issues need to be addressed before the manuscript can be considered for publication.
- The study relies primarily on GWAS data from European populations. This limits the generalizability of the findings to other ethnic groups. It is crucial to include multi-ethnic GWAS datasets to enhance the applicability of the genetic causal inference. The authors should discuss the potential impact of this limitation on their conclusions and consider incorporating more diverse datasets in future analyses.
- The sample size of the CeD dataset (GSE112102) is relatively small, which may affect the robustness of the transcriptomic findings. The authors should acknowledge this limitation and consider integrating additional large-scale datasets to validate their results further. A larger sample size would provide more reliable insights into the shared molecular pathways.
- While the authors identified P2RY8, ITGAL, and GPR65 as hub genes through bioinformatics analysis, functional studies are still needed to elucidate their precise roles in immune dysregulation. The authors should include gene silencing and overexpression experiments to demonstrate the functional impact of these genes on disease mechanisms. This would strengthen the biological relevance of their findings.
- Although the authors performed various sensitivity analyses to assess the robustness of their Mendelian randomization results, they should provide more detailed information on the potential biases and assumptions underlying their analysis. This includes discussing the potential impact of pleiotropy and the choice of instrumental variables on their causal inferences.
- The molecular docking analysis suggests that Trichostatin A binds to the identified hub genes. However, the authors should provide more detailed discussion on the biological implications of these binding interactions. This includes exploring the potential downstream effects of Trichostatin A on immune signaling pathways and its therapeutic potential in the context of these diseases.
- The authors should provide a more comprehensive comparison of their findings with existing studies on the shared mechanisms of CD, CeD, and AS. This would help contextualize their results within the broader field and highlight the novelty of their contributions.
Author Response
Comment 1:
The study relies primarily on GWAS data from European populations. This limits the generalizability of the findings to other ethnic groups. It is crucial to include multi-ethnic GWAS datasets to enhance the applicability of the genetic causal inference. The authors should discuss the potential impact of this limitation on their conclusions and consider incorporating more diverse datasets in future analyses.
Response:
Thank you for highlighting this important point. We agree that our study’s reliance on GWAS data from European populations may limit the generalizability of the findings to other ethnic groups. We have now explicitly acknowledged this limitation in the revised Discussion section(lines:534-539) and emphasized the need for future studies incorporating multi-ethnic GWAS datasets to improve the applicability of genetic causal inferences across diverse populations.
Comment 2:
The sample size of the CeD dataset (GSE112102) is relatively small, which may affect the robustness of the transcriptomic findings. The authors should acknowledge this limitation and consider integrating additional large-scale datasets to validate their results further. A larger sample size would provide more reliable insights into the shared molecular pathways.
Response:
Thank you for the reviewer’s insightful comment. We acknowledge that the relatively small sample size of the CeD dataset (GSE112102) is a limitation that may affect the robustness of our transcriptomic findings. However, we selected this dataset because it is one of the few publicly available transcriptomic datasets specifically focused on CeD, and it remains a valuable resource for exploring disease-specific molecular signatures. We have clarified this point in the revised Discussion section(lines:542-547). In addition, we emphasized the need for future studies to incorporate larger and more diverse transcriptomic datasets to validate and extend our findings.
Comment 3:
While the authors identified P2RY8, ITGAL, and GPR65 as hub genes through bioinformatics analysis, functional studies are still needed to elucidate their precise roles in immune dysregulation. The authors should include gene silencing and overexpression experiments to demonstrate the functional impact of these genes on disease mechanisms. This would strengthen the biological relevance of their findings.
Response:
Thank you for this important and insightful suggestion. We fully agree that functional validation experiments, such as gene silencing and overexpression, are highly valuable for clarifying the biological roles of P2RY8, ITGAL, and GPR65 in immune dysregulation. These studies would indeed strengthen the biological relevance of our findings and provide more direct evidence of their functional impact.
We sincerely apologize for not being able to conduct these experiments within the current study, due to time constraints and the scope of the project. The primary aim of this work was to explore the shared molecular mechanisms underlying Crohn’s disease, celiac disease, and ankylosing spondylitis through integrative bioinformatics analyses, including genetic and transcriptomic approaches. Nevertheless, we believe that the current bioinformatics findings provide a meaningful and coherent basis for future mechanistic investigations. We have explicitly acknowledged this limitation in the revised Discussion section(lines:548-559) and have planned to conduct targeted functional experiments in follow-up studies to further explore the roles of these hub genes in disease pathogenesis.
Comment 4:
Although the authors performed various sensitivity analyses to assess the robustness of their Mendelian randomization results, they should provide more detailed information on the potential biases and assumptions underlying their analysis. This includes discussing the potential impact of pleiotropy and the choice of instrumental variables on their causal inferences.
Response:
Thank you for this important comment. We fully acknowledge the importance of addressing the underlying assumptions of Mendelian randomization (MR) analysis, including potential biases such as horizontal pleiotropy and the selection of instrumental variables. In our study, we conducted multiple sensitivity analyses (e.g., MR-Egger, MR-PRESSO, weighted median) to test the robustness of our results and to account for possible pleiotropy. We also carefully selected instrumental variables based on strict criteria, including genome-wide significance (p < 5×10⁻⁸), independence (clumping with r² < 0.01), and biological plausibility. Due to space limitations, we were unable to include a more detailed discussion of MR assumptions in the current manuscript. However, we agree that this is an important aspect and plan to elaborate on it in future revisions or follow-up publications. We sincerely appreciate the reviewer’s helpful suggestions.
Comment 5:
The molecular docking analysis suggests that Trichostatin A binds to the identified hub genes. However, the authors should provide more detailed discussion on the biological implications of these binding interactions. This includes exploring the potential downstream effects of Trichostatin A on immune signaling pathways and its therapeutic potential in the context of these diseases.
Response:
Thank you for this insightful suggestion. We have expanded the Discussion section to explore the potential downstream effects of Trichostatin A on immune signaling pathways(lines:515-528), such as its role in modulating histone acetylation, regulating gene expression, and impacting pro-inflammatory cytokine production. We also discussed its potential therapeutic implications in the context of CD, CeD, and AS, highlighting its relevance as a candidate for repurposing in immune-mediated diseases.
Comment 6:
The authors should provide a more comprehensive comparison of their findings with existing studies on the shared mechanisms of CD, CeD, and AS. This would help contextualize their results within the broader field and highlight the novelty of their contributions.
Response:
Thank you for this helpful suggestion. We have expanded both the Discussion(lines:502-507) and Introduction sections(lines:44-46) to provide a more comprehensive comparison of our findings with previous studies on the shared mechanisms of CD, CeD, and AS. We discussed how our results align with or differ from prior work, and how our integrative approach—combining Mendelian randomization, transcriptomic analysis, and molecular docking—offers new insights that complement existing knowledge. These additions help contextualize our results within the broader field and highlight the novel aspects of our contributions.
Round 2
Reviewer 2 Report
Comments and Suggestions for Authors
Thank you for the conducted corrections.
Author Response
We sincerely thank the reviewer for taking the time to evaluate our revised manuscript. We greatly appreciate your recognition of the corrections, and we are grateful for your constructive comments, which helped improve the quality and clarity of our work.
Reviewer 3 Report
Comments and Suggestions for Authors The authors addressed all the questions and comments and significantly improved the manuscripts with appropriate corrections.Author Response
We are grateful for your encouraging feedback. We appreciate the opportunity to revise our manuscript in response to your insightful suggestions, which enabled us to clarify our study’s positioning and strengthen the scientific presentation. Thank you again for your time and valuable input.